# Characterization of GafChromic EBT2 film dose measurements using a tissue-equivalent water phantom for a Theratron® Equinox Cobalt-60 teletherapy machine

**Daniel Akwei Addo** [1]*, **Elsie Effah Kaufmann** [2], **Samuel Nii Tagoe**[3,4], **Augustine Kwame Kyere**[5]

**1** Department of Computer Engineering, Kwame Nkrumah University of Science and Technology, Kumasi, Ghana, **2** Department of Biomedical Engineering, School of Engineering Sciences, University of Ghana, Legon, Accra, Ghana, **3** National Radiotherapy Oncology and Nuclear Medicine Centre, Korle-Bu, Accra, Ghana, **4** School of Biomedical and Allied health Sciences, University of Ghana, Accra, Ghana, **5** Medical Physics Department, Graduate School of Nuclear and Allied Sciences, University of Ghana, Atomic, Accra, Ghana

\* danieladdo@knust.edu.gh

**Data Availability Statement:** All relevant data are within the manuscript and its Supporting information files.

## Abstract

### Purpose

In vivo dosimetry is a quality assurance tool that provides post-treatment measurement of the absorbed dose as delivered to the patient. This dosimetry compares the prescribed and measured dose delivered to the target volume. In this study, a tissue-equivalent water phantom provided the simulation of the human environment. The skin and entrance doses were measured using GafChromic EBT2 film for a Theratron® Equinox Cobalt-60 teletherapy machine.

### Methods

We examined the behaviors of unencapsulated films and custom-made film encapsulation. Films were cut to 1 cm × 1 cm, calibrated, and used to assess skin dose depositions and entrance dose. We examined the response of the film for variations in field size, source to skin distance (SSD), gantry angle and wedge angle.

### Results

The estimated uncertainty in EBT2 film for absorbed dose measurement in phantom was ±1.72%. Comparison of the measurements of the two film configurations for the various irradiation parameters were field size ($p = 0.0193$, $\alpha = 0.05$, $n = 11$), gantry angle ($p = 0.0018$, $\alpha = 0.05$, $n = 24$), SSD ($p = 0.1802$, $\alpha = 0.05$, $n = 11$) and wedge angle ($p = 0.6834$, $\alpha = 0.05$, $n = 4$). For a prescribed dose of 200 cGy and at reference conditions (open field 10 cm x 10 cm, SSD = 100 cm, and gantry angle = 0º), the measured skin dose using the encapsulation material was 70% while that measured with the unencapsulated film was 24%. At reference irradiation conditions, the measured skin dose using the unencapsulated film was higher for

**Funding:** The author(s) received no specific funding for this work.

**Competing interests:** The authors have declared that no competing interests exist.

open field configurations (24%) than wedged field configurations (19%). Estimation of the entrance dose using the unencapsulated film was within 3% of the prescribed dose.

## Conclusions

GafChromic EBT2 film measurements were significantly affected at larger field sizes and gantry angles. Furthermore, we determined a high accuracy in entrance dose estimations using the film.

## Introduction

Several types of external photon beam radiotherapy exist [1–5]. Compared to fixed beam radiotherapy (where there is no modulation in the beam delivered), with intensity-modulated radiotherapy, modulation of the photon beam involves using multi-leafs collimator and compensators [3, 6, 7]. Image-guided radiotherapy is another type of external beam radiotherapy; however, with this technique, the treatment is guided by images obtained from Computed Tomography or Magnetic Resonance Imaging [5, 8–10]. The radiation treatment process requires the control and assurance of quality in the overall treatment delivery. Several tools are used to ensure the radiotherapy process is safe and effective. These include port films [11, 12]; treatment planning systems (TPS) [13, 14]; and phantoms (used to simulate the human environment in the provision of scatter) [15–17]. These traditional quality assurance tools are used before radiotherapy, complicating the detection and rectification of treatment errors [18–20].

*In vivo* dosimetry is another method for assuring quality in external photon beam radiotherapy and is performed whilst the patient is receiving treatment [21, 22]. Dosimeters placed in body cavities or on the patient's skin measure the absorbed doses. *In vivo* dosimetry may be employed to detect errors in individual patients [23, 24], assess errors in core procedures [25], evaluate the quality of specific treatment techniques [26] or quantify the dose in cases where the dose estimation is inaccurate or difficult to calculate [27]. This mode of dosimetry involves the measurement of two types of absorbed doses. The entrance dose is quantified at the entrance of the beam while the exit dose refers to the radiation dose estimated at the exit point of the photon beam.

Different types of *in vivo* dosimeters are used in external photon beam radiotherapy. In this research, GafChromic® EBT2 film (produced by International Specialty Products, NJ, USA) was used because of the following characteristics: ease of handling, thin width, good spatial resolution, self-developing, nearly tissue equivalent characteristics (with an effective atomic number of 6.84) [28–30], and wide dose range (between 0.01 and 10 Gy) [29, 31, 32]. After the film is irradiated, the active component in the film undergoes a polymerization process resulting in a color change to blue. Changes in the optical density of the film (resulting from the polymerization process) are used for absorbed dose quantification.

Dose measurements by EBT2 films are affected by contaminating electrons [33–35]. Several factors contributing to electron contamination have been investigated in previous studies [36–39]. Medina, Teijeiro [39] quantified electron contamination and reported that the degree of electron contamination depends on factors such as photon energy, beam shaper, treatment field and SSD. They further reported a greater dependence of electron contamination on field size, SSD, and depth for an 18 MV photon energy when compared with a 6 MV photon [39]. Klevenhagen [38] studied the implication of electron backscatter on electron dosimetry. Their study concluded that decreasing the treatment field size resulted in a surface dose reduction.

Because of the numerous factors causing electron contamination for film dosimetry, estimation of correction factors can be laborious, and this has the potential of negatively impacting the accuracy of dose measurements. In this study, we developed a new film setup configuration whereby the film is "housed" within an encapsulation material with the aim of minimizing electron contamination. The effects of such film configuration on dose estimations have not yet been studied. A tissue equivalent water phantom was used in this research to simulate the human environment. The general aim of this study was to characterize GafChromic EBT2 film dose measurements using a tissue-equivalent water phantom for a Theratron® Equinox Cobalt-60 Teletherapy Machine. The specific questions that this study sought to address were:

1. How do the two configurations of the film compare in terms of their response to variations in irradiation parameters?

2. How do variations in irradiation parameters affect skin dose?

3. How accurate are entrance dose calculations by these films?

## Materials and methods

We evaluated the behaviors of irradiated films using Theratron® Equinox-100 Cobalt-60 teletherapy unit with an average photon energy of 1.25 MeV for the following irradiation parameters: field size, SSD, gantry angle and wedge angle. The film used in this study was manufactured by International Specialty Product inc. with Lot number A06271203. According to the film's specifications, the EBT2 film can measure absorbed doses between 1 cGy and 10 Gy. The dimension of the film sheets was 8 cm × 10 cm. We cut each film sheet into 1 cm × 1 cm for calibration, measurement of skin dose and entrance dose estimations. The beam output from the Equinox-100 Cobalt-60 teletherapy unit was measured using a mini water phantom (with the dimension 20 cm x 20 cm x 20 cm). The water phantom was made of Perspex and can accommodate a 0.6 cm$^3$ farmer type ionization chamber. The ionization chamber used was a cylindrical farmer chamber type (model PTW30001) manufactured by PTW Freiburg with serial number 1510 and a volume of 0.6 cm$^3$.

We determined the dose rate of the Cobalt-60 unit regularly at reference conditions of field size (10 cm × 10 cm), SSD (100 cm) and gantry angle (0˚). These reference conditions were kept constant throughout this study. The calibration of the ionization chamber is traced to the specifications of the International Atomic Energy Agency's secondary standard laboratory. The ionization chamber's calibration factor (ND, W) was 5.17 and determined for the following: a chamber bias voltage of +400 V, the temperature of 20˚C and pressure of 101.325 kPa for humidity not exceeding 70%. We used the ionization chamber to establish a dosimetric protocol for the GafChromic film. An electrometer (model PTW UNIDOS with Serial Number T10005) connected to the ionization chamber was used to quantify the charges created by ionization. Both the electrometer and ionization chamber were calibrated together for the dosimetric procedures. The densitometer used for measuring the optical densities of the films was manufactured by PTW Freiburg (Germany), called DensiX, and had a serial number of T52001–3263.

### Film preparation

We handled the films according to accredited international protocols [29, 40, 41]. To use EBT2 films for *in vivo* dosimetry, they were cut into squares with the dimension of 1 cm × 1 cm. The films were cut with precaution to minimize scratches and oiling. The same faces of

the radiochromic films were marked (because of the asymmetry in the film's structure) to ensure consistency during radiation exposures. The films were kept in a dark-airtight box to minimize unnecessary exposure to light and moisture and later used for assessing the effects of irradiation parameters on skin dose and entrance dose estimations. For each film, the optical density before irradiation was measured (and recorded as *OD1*) to correct for background radiations. The point densitometer was warmed for 5 minutes before being used to increase the accuracy of the measurements.

## Postexposure optical density growth of EBT2 film

Previous studies have shown that optical densities of irradiated EBT2 increase with time [42, 43]. We performed a postexposure optical density growth experimentation to estimate the optimum time for measuring the optical densities of the irradiated EBT2 films. Four sets of films, each consisting of three films, were irradiated to doses of 50 cGy, 200 cGy, 400 cGy and 800 cGy. The irradiated films' optical densities were then measured with the point densitometer at intervals of 60 minutes between 1 minute and 6480 minutes.

## Calibration curve for EBT2 film

When radiochromic films are exposed to ionization radiations, the absorbed dose is expressed through changes in optical density. In radiation dosimetry, it is the absorbed dose which is required. The changes in optical density are converted to absorbed dose through calibrations. The films were calibrated using the International Atomic Energy Agency (IAEA) TRS398 protocol [44, 45]. We placed ten sets of films between solid water slabs with the dimension 30 cm × 30 cm × 25 cm and the slabs positioned perpendicularly to the direction of the beam. The films were at a depth of 5 cm beneath the surface of the slabs. At reference conditions, the sets of films were irradiated at different treatment times to give absorbed doses of 50 cGy, 100 cGy, 150 cGy, 200 cGy, 250 cGy, 300 cGy, 350 cGy, 400 cGy, 450 cGy and 800 cGy. After irradiating the films, they were scanned using the densitometer, and their optical density was recorded as *OD2*. We used Eq 1 in estimating the net optical density (*NOD*). The absorbed doses determined by ionization chamber readings were plotted against their corresponding net optical densities to obtain the calibration curve. The calibration curve was subsequently used to convert any film signal (optical density) to their absorbed dose.

$$NOD = OD2 - OD1 \tag{1}$$

Where: *OD1* is the optical density of the film before irradiation.

## Dose uncertainty budget

The dose uncertainties carried out in this study were calculated by error propagation as used in previous studies [46–48]. Two sources of uncertainties considered during the generation of the calibration curve were: the experimental and fitting. We assumed the scanning of the films to be the source of the experimental uncertainties, while the curve fitting uncertainties were associated with the accuracy of the curve fitting process.

Eq 2 represents a potential equation for the calibration curve:

$$Dose = aOD^3 + bOD^2 + cOD + d \tag{2}$$

Where:
*OD* is the measured optical density; *Dose* is the estimated dose; *a*, *b*, *c* and *d* are the coefficients.

The final experimental dose uncertainty was calculated by applying Eq 3:

$$\delta_{e-dose} = (3aOD^2 + 2bOD + c) * \delta_{OD} \tag{3}$$

where: $\delta_{e-dose}$, is the experimental dose uncertainty; $OD$, is the optical density measurement; $\delta_{OD}$, is the standard deviation of the densitometer. Since the digital display of the densitometer is limited to 2 decimal places, the standard deviation of the densitometer was approximated to 0.01.

The curve fitting uncertainty was calculated by applying Eq 4:

$$\delta_{f-dose} = \sqrt{\delta_a^2 OD^6 + \delta_b^2 OD^4 + \delta_c^2 OD^2 + \delta_d^2)} \tag{4}$$

Where: $\delta_{f-dose}$ is the fitting uncertainty; $\delta_a$, $\delta_b$, $\delta_c$, & $\delta_d$ are the standard deviation of the fitting parameters. The standard deviation of the fitting parameters was calculated using the non-linear model fit function in MATLAB (The Math Works, Inc. *MATLAB*. Version 2020a).

Finally, the total dose uncertainty was calculated using the Eq 5:

$$\delta_{dose} = \sqrt{(\delta_{e-dose}^2 + \delta_{f-dose}^2)} \tag{5}$$

## Design of encapsulation cap

We constructed the encapsulation medium (for holding the EBT2 film) using Perspex. To ensure that the film was measuring the absorbed dose at a depth of electronic equilibrium (Dmax), a 0.5 cm thickness of encapsulation material was placed above the film. The cross-sectional diameter of the encapsulation material used in the experimentation phase was 4 cm for easy handling. This diameter is expected to be reduced in actual practice to minimize the encapsulation material interference in the overall treatment delivery. Fig 1 shows the constructed encapsulation cap used in this study.

## Entrance dose calibration factor of EBT2 film

Because the absorbed dose measured by EBT2 film is not at Dmax, which ideally quantifies the entrance dose, calibration of the film was performed for entrance dose measurements. We determined the entrance dose calibration factor by positioning and irradiating the radiochromic films (at reference conditions) on the surface of a water phantom. Fig 2 shows the setup for the entrance dose estimations. As illustrated in Fig 2, the ionization chamber was positioned along the central axis at a distance of 0.5 cm. At this depth beneath the surface of the phantom, the ionization chamber was probing the percentage depth dose curve at its maximum dose and not its subsequent fall-off.

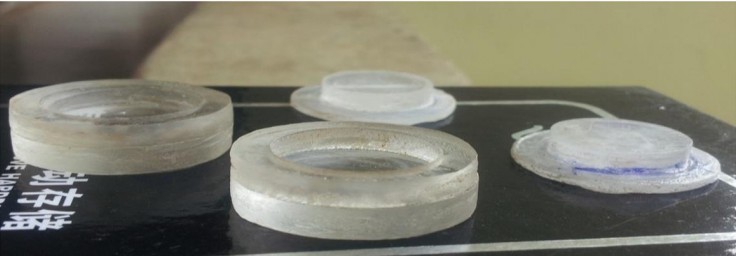

**Fig 1. Encapsulation cup used for holding the films for radiation dosimetry.** The films were positioned at Dmax. The cup was constructed with Perspex and had a dimension of 4 cm (cross-sectional diameter) x 1 cm (thickness).

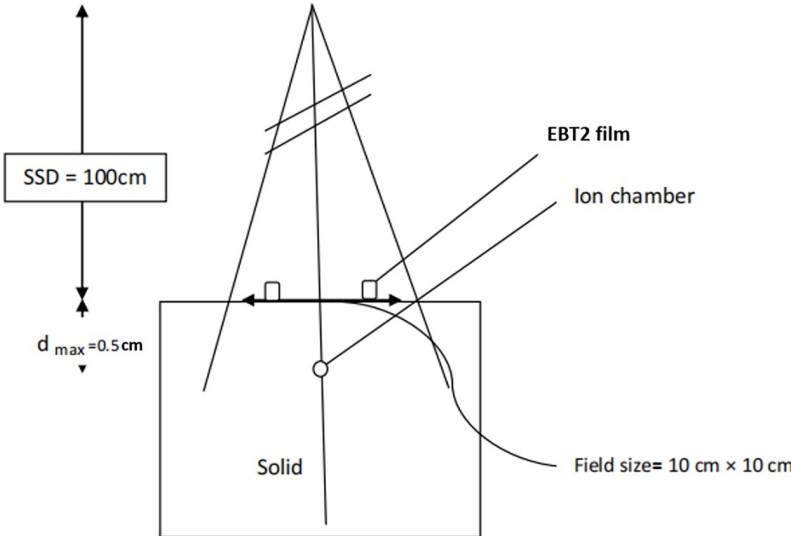

**Fig 2. Schematic diagram showing how the calibration of EBT2 film was performed.** The films were placed on the surface of the water phantom, while the ionization chamber was placed at Dmax. The irradiation was performed at reference conditions.

In determining the entrance dose calibration factor, Eq 6 was used.

$$Fcal, entrance = \frac{(Ric)\, reference\ conditions}{(Rf)\, reference\ conditions} \tag{6}$$

Where: *Fcal*, *entrance* is the entrance dose calibration factor
*Ric* is the reading of the ionization chamber at reference condition
*Rf* is the reading of EBT2 at reference condition

## Correction factors determination when using EBT2 films

Absorbed dose measurements using EBT2 film are highly influenced by contaminating electrons from the air and head of the treatment machine [49, 50]. The proportions of these contaminating electrons vary depending on the irradiation conditions, hence the need for correction factors to minimize these interferences and enhance the accuracy of measurements. In determining the correction factors, we irradiated the films for varying field sizes (4 cm x 4 cm to 24 cm x 24 cm), SSD (75 cm to 120 cm), gantry angle (0º to 90º) and wedge angle (15º to 60º). When assessing the effect of each irradiation parameter on the film response, we kept the other irradiation parameters at reference conditions.

For the same irradiation condition (as performed on the films), the ionization chamber was used to measure the dose delivered to the water phantom. The data obtained from the irradiations (both the film and ionization chamber) were normalized with measurements obtained at reference conditions. I.e., The correction factor for each irradiation parameter was calculated by dividing the ratio of the ionization chamber's reading to the film's reading under clinical conditions by the same ratio under reference irradiation conditions. The equation for

estimating the correction factors is shown in Eq 7.

$$CF = \frac{\left(\frac{Ric}{Rf}\right) clinical\ condition}{\left(\frac{Ric}{Rf}\right) reference\ condition}$$ (7)

Where: *CF* is the correction factor
*Ric* is the reading of the ionization chamber
*Rf* is the reading of the EBT2 film

## Entrance dose calculations

We estimated the entrance dose using the calibration factor, correction factors and the film's reading. The entrance doses were determined using Eq 8. The absorbed radiation dose at any depth within the tissue was determined by multiplying the entrance dose by the percentage depth dose (PDD) for Cobalt-60 energy.

$$Dentrance = Df * Fcal, entrance * \prod CF$$ (8)

Where: *Dentrance* represents the entrance dose and is the dose at Dmax
*Df* represents the EBT2 film dose reading
*Fcal*, entrance represents the entrance dose calibration factor
Π*CF* represents the product of correction factors (for field size, SSD, gantry angle and wedge angle) used during the dose estimations

## Skin dose determination

The skin dose associated with radiotherapy is of interest for clinical evaluation or examination of the risk of late effects of radiation. The skin dose estimated at a depth of 0.070 mm measures the amount of radiation deposited in the skin tissue. We quantified the skin dose using two film configurations (an unencapsulated film and an encapsulated film). For the encapsulated film configuration, skin dose was determined by placing the film between the encapsulation material and the skin. Because the depth of the active layer within the film (depth of 0.080 mm) was slightly higher than the required depth for skin dose estimation (0.070 mm), a calibration factor was determined and used to convert the absorbed dose by the film to that of the skin. In estimating the calibration factor for skin dose estimations, a single EBT2 film was irradiated at reference conditions. The optical density of the film was then measured and recorded. Four films were also irradiated under the same reference conditions. The optical density of the bottom-placed film was then measured and recorded. The depth of the active layer of the bottom-placed film was 0.935 mm. The optical density at a depth of 0.070 mm was estimated using a simple extrapolation technique from the measured optical density for both the single film and four film setups. The steps involved in determining the skin dose calibration factor, skin dose and percentage skin dose are shown below.

$$Dskin = Df \times Fcal, skin$$ (9)

Where: *Df* is the film's reading.

$$Fcal, skin = \frac{estimated\ film\ dose\ at\ a\ depth\ of\ 0.07}{estimated\ film\ dose\ at\ a\ depth\ of\ 0.08}$$ (10)

In determining the dose at a depth of 0.07 mm within the film, the extrapolation method

shown below was used.

$$0.080 \, mm \equiv 0.03 \, (optical \, density)$$

$$0.935 \, mm \equiv 0.08 \, (optical \, density)$$

$$0.070 \, mm \equiv X \, (optical \, density)$$

$$\text{Therefore} : \frac{(0.935) - (0.080)}{(0.935) - 0.070} \equiv \frac{(0.08) - (0.03)}{(0.08) - x}$$

$X$ was estimated to be 0.03

$$\text{Therefore } X = 0.03 \, (optical \, density)$$

$$Fcal, \; skin = \frac{0.03}{0.03} = 1.00$$

$$Dose \; to \; skin \, (Dskin) = Df * 1$$

$$\% \; skin \; dose = \frac{Dskin}{Dentrance} * 100 = \frac{Df * 1.00}{Df * Fcal, \; entrance * \prod CF} * 100$$

$$\% \; skin \; dose = \frac{1.00}{Fcal, \; entrance * \prod CF} * 100$$

But $Fcal$, $entrance$ for unencapsulated film was calculated to be 4.134.

$$\text{Therefore} : \; \% \; skin \; dose = \frac{1.00}{4.134 * \prod CF} * 100 \approx \frac{24}{\prod CF} \tag{11}$$

## Results

### Post-irradiation optical density growth

From Fig 3, after irradiating the films, their optical densities increased and gradually became asymptotic to the horizontal axis with time. Films irradiated with higher doses of radiation recorded higher optical densities than those exposed to lower doses. Low doses of radiation (50 cGy, 200cGy) caused the optical density growth to stabilize quickly compared to the films irradiated with higher doses (400 cGy, 800 cGy).

When the post-exposure optical density growth for an absorbed dose of 50 cGy was compared with that of a 200 cGy absorbed dose, the difference was statistically significant ($p < 0.001$, $\alpha = 0.05$, n = 31). Comparing the optical density growth for a 200 cGy absorbed dose to a 400 cGy absorbed dose showed a statistically significant difference between the two datasets ($p < 0.001$, $\alpha = 0.05$, n = 31). Furthermore, when the optical density growth for a 400 cGy absorbed dose was compared to that of an 800 cGy absorbed dose, a statistically significant difference was observed ($p < 0.001$, $\alpha = 0.05$, n = 31). From the results, it was observed that after 24 hours (1440 minutes) of exposure, the optical densities of all the exposed films were stabilized.

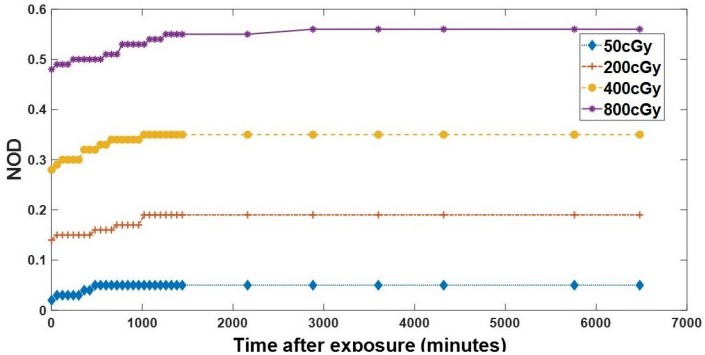

**Fig 3. Post-irradiation optical density growth of EBT2 films.** As the time after the films' irradiation increased, a corresponding increase in the optical density growth is observed. At longer post-exposure times, the optical density growth stabilized.

## Calibration curve for EBT2 film

The calibration curve for the EBT2 film is shown in Fig 4 and was obtained by plotting absorbed dose measurements obtained from the ionization chamber against the NOD of EBT2 films. Fig 4 shows that as the absorbed dose increased, the NOD of EBT2 films also increased. A 3rd-degree polynomial represented the line of best fit that relates the net optical density to absorbed dose and is shown in Eq 12. The correlation coefficient for the plot was estimated to be 0.9989. The calibration curve was used to convert optical density measurements to the absorbed dose.

$$y = 1685X^3 - 187.9X^2 + 1006X - 2.328 \qquad (12)$$

Where $X$ represents the optical density and y the absorbed dose in gray.

For the same batch number of films and irradiation conditions, the characteristics of the calibration curve may change depending on the densitometer or scanner used. Hence, the calibration curve should be re-evaluated when a different densitometer or scanner is used for the optical density measurements.

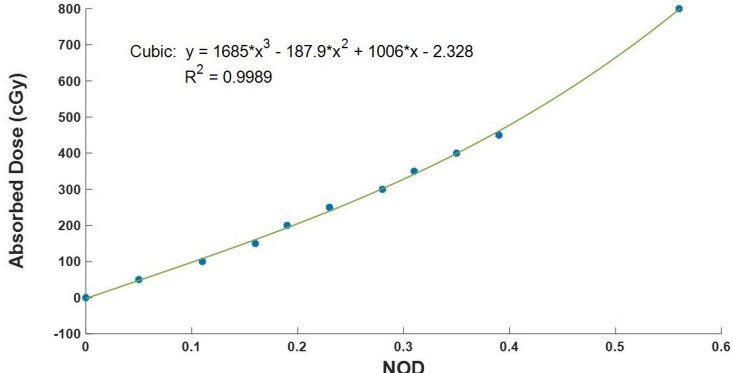

**Fig 4. Calibration curve for converting the film optical density readings to absorbed dose in gray.** The absorbed dose measurement by the ionization chamber was plotted against the NOD of the films.

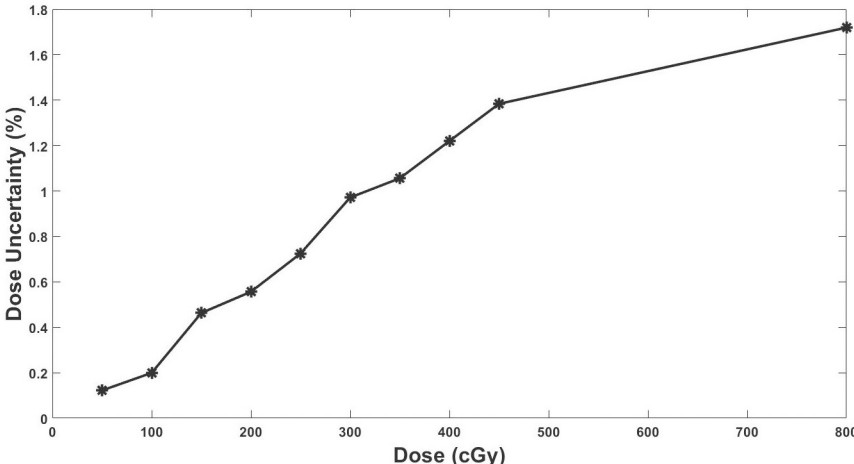

**Fig 5. Plot of total dose uncertainty against absorbed dose for EBT2 film.** The percentage uncertainty in absorbed dose estimations increased as the dose to the film increased.

## Dose uncertainty budget

Fig 5 shows a plot of dose uncertainties against estimated absorbed doses using EBT2 films. From Fig 5, increasing the absorbed radiation dose increased the absorbed dose uncertainty. The minimum dose uncertainty recorded was 0.12% (at 50cGy), while the maximum was 1.72% (at 800 cGy).

## Assessing the effects of irradiation parameters

Fig 6 shows the effects of irradiation parameters on optical density measurements by the unencapsulated and encapsulated film. The results in Fig 6 were normalized under reference conditions (10 cm x10 cm; SSD = 100 cm; Gantry angle 0˚). From Fig 6A, there was a general increase in the normalized optical density (ND) of the unencapsulated film compared to the encapsulated film when field size increased. The increase in ND was due to the increasing number of contaminating electrons from the collimator and air with an increase in field size. Because of the low penetrating power of the contaminating electrons, they were absorbed by the unencapsulated film. Changes in the field size had a relatively negligible effect on the encapsulated film because the encapsulation material absorbed most of the contaminating electrons. The difference in the response of encapsulated and unencapsulated film for variations in field size was statistically significant *(p = 0.0193, α = 0.05, n = 11)*.

From Fig 6B, as SSD increased, the optical density of the films (both unencapsulated and encapsulated) decreased due to the decrease in the dose rate of the Equinox 100 Co-60 machine with increasing SSD. At shorter SSD, the films were exposed to large quantities of low-energy photons scattered by components in the Co-60 teletherapy unit, inducing a slight over-response of the film. At large SSD, this contamination was insufficient to contribute to the absorbed dose. For changes in SSD, ND for the encapsulated films was statistically not different from that of the unencapsulated films *(p = 0.1802, α = 0.05, n = 11)*.

Fig 6C shows the effects of gantry angle on the measurements of the unencapsulated and encapsulated films. From Fig 6C, the response of the encapsulated EBT2 film with gantry angle was almost constant. However, there was a general increase in the optical density measurements for the unencapsulated film when the gantry angle increased due to the shift of

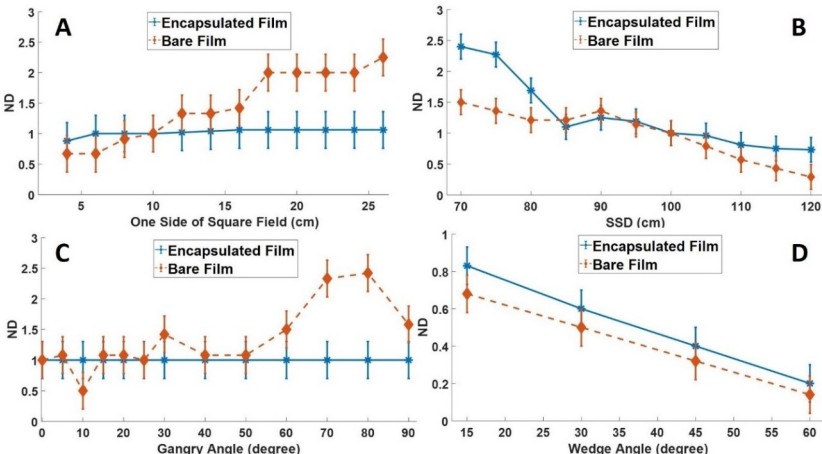

**Fig 6. Effects of irradiation parameters on the optical density measurements of unencapsulated and encapsulated film.** The results in Fig 6 were normalized to reference conditions (10 cm x10 cm; SSD = 100 cm; Gantry angle 0˚). A: Shows the effect of field size, B: Shows the effect of SSD, C: Shows the effect of gantry angle, D: Shows the effect of wedge angle.

contaminating electrons onto the film. Maximum film readings for the unencapsulated film occurred at gantry angles between 70º and 80º. A statistically significant difference was obtained when we compared the ND of the unencapsulated film with the encapsulated film ($p = 0.0018$, $\alpha = 0.05$, $n = 24$).

From Fig 6D, at a reference field size of 10 cm × 10 cm, an increase in the wedge angle decreased ND for both the unencapsulated and encapsulated film. The decrease in ND was because as the wedge angle increases, the thickness of attenuating material increases, causing an increase in the average energy of the primary and secondary photons. Comparing the ND of the encapsulated film with the unencapsulated film did not result in a statistically significant difference ($p = 0.6834$, $\alpha = 0.05$, $n = 4$).

## Skin dose assessment

At reference irradiation conditions, we assessed the impact of encapsulation material on skin dose for a prescribed absorbed dose of 200 cGy to Dmax.

a) Skin dose estimation without encapsulation material

$$Dskin = Df \times Fcal, skin$$

$$Df = 0.05 \equiv 48.7104 cGy$$

$$Fcal, skin = 1.00$$

$$\text{Therefore}: Dskin = 48.7104 \times 1.00 = 48.71 cGy$$

$$\%skin\,dose = \frac{48.71}{200.00} * 100$$

$$\%\,Skin\,dose \approx 24\%$$

b) Skin dose estimation with encapsulation material

$$Dskin = Df \times Fcal, skin$$

$$Df = 0.14 \equiv 140.8152 cGy$$

$$Fcal, skin = 1.00$$

$$\text{Therefore}: Dskin = 140.08125 \times 1.00 = 140.08 cGy$$

$$\%skin\ dose = \frac{140.08}{200.00} * 100$$

$$\%\ Skin\ dose \approx 70\%$$

From the preliminary percentage skin dose estimation for the unencapsulated and encapsulated film, the encapsulation around the film increased the skin dose by approximately three times under reference conditions. Only the unencapsulated films were used for the subsequent skin dose measurements due to the high skin dose associated with using encapsulated films.

In Fig 7, the effects of irradiation parameters on skin dose depositions are assessed under reference irradiation conditions. Increasing the field size increased the skin dose due to the increase in contaminating electrons. As SSD increased, the deposited skin dose decreased because only electrons or photons with sufficient energy interacted with the skin. Large gantry angles had relatively high values in skin dose due to the shift of the contaminating electron region towards the skin. Furthermore, increasing the wedge angles decreased the skin dose due to increased attenuation material volume. The skin dose observed with wedged fields was relatively lower than those of open irradiation fields. For example, at reference conditions of field size, SSD, and gantry angle, using a 45º wedge filter reduced the skin dose by a factor of two compared to an open field.

## Entrance dose assessment

Fig 8 shows the percentage accuracy in entrance dose estimation for variations in field size, SSD, gantry angle and wedge angle. The percentage accuracy in entrance dose measurements was calculated by comparing the estimated entrance dose using the film with the prescribed dose. As shown in Fig 8A, for field size ranging from 4 cm x 4 cm to 24 cm x 24 cm, the maximum and minimum percentage entrance dose difference were 2.1% (at field size of 6cm x 6 cm and 12 cm x 12 cm) and 0.3% (at field size of 18 cm x 18 cm). From Fig 8B, variations in the SSD resulted in a maximum percentage entrance dose of 2.3% at SSD of 90 cm and a minimum percentage entrance dose of 0.4% at SSD of 75 cm. As illustrated in Fig 8C, the maximum and minimum percentage entrance dose differences were 2.32% (at a gantry angle of 70º) and 0.03% (at a gantry angle of 10º) for gantry angles ranging from 0º to 90º. Furthermore, from Fig 8D, for field sizes ranging from 4 cm x 4 cm to 18 cm x 18 cm, when the wedge angle changed from 15º to 60º, the maximum percentage entranced dose difference was 1.72% (at a field size of 4 cm x 4 cm and wedge angle of 45º). The minimum percentage entrance dose difference was 0.01% at a field size of 12 cm x 12 cm and a wedge angle of 60º.

## Discussion

Radiation therapy involves the delivery of a prescribed dose to the target volume, sparing the surrounding healthy organs and tissues as much as possible. Traditional techniques that ensure

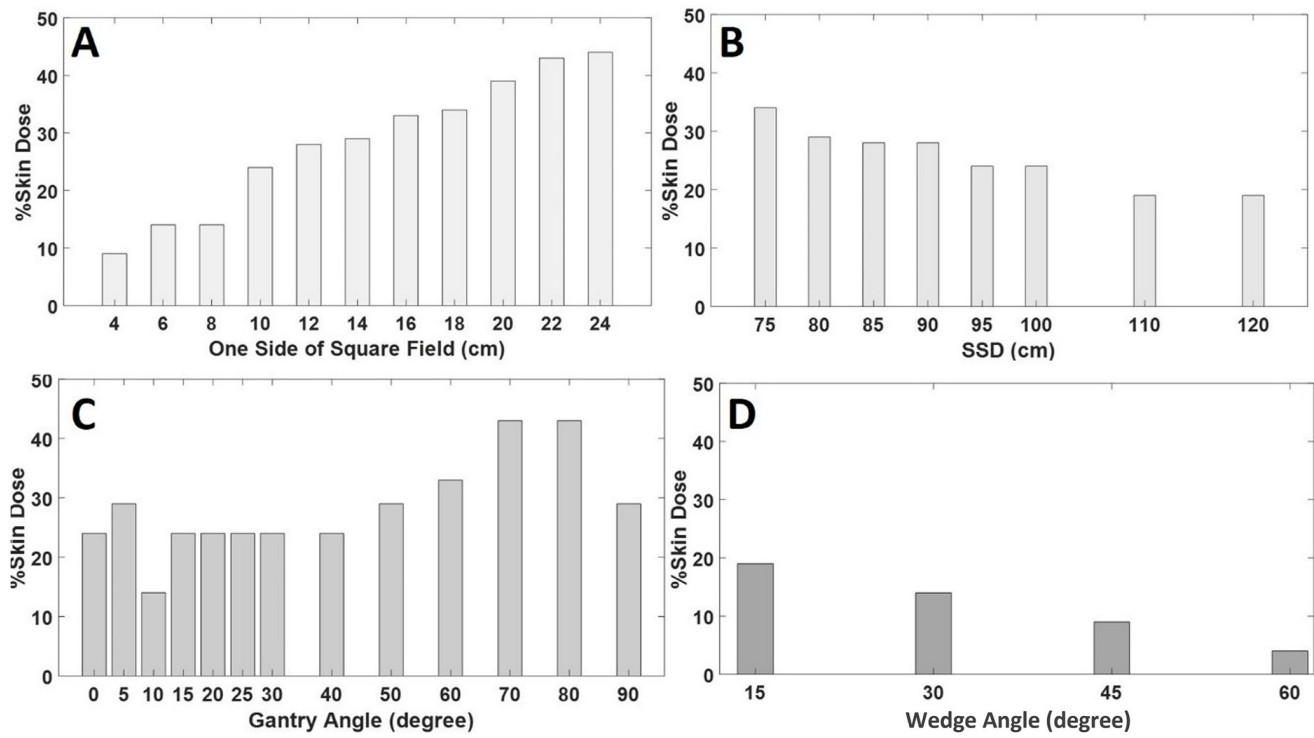

**Fig 7. Effects of irradiation parameters on skin dose deposition.** The results of the skin dose depositions were obtained for a reference dose of 200cGy A: Effect of Field size; B: Effect of SSD; C: Effect of Gantry angle; D: Effect of Wedge angle.

the prescribed dose gets delivered to the target volume include treatment planning systems and phantom studies. These dose monitoring methods are done before the radiation treatment and may not reflect the radiation delivered during treatment. *In vivo* dosimetry is used during radiotherapy to assess the real-time absorbed dose to the target volume. There are several *in vivo* dosimeters available; however, GafChromic EBT2 film was used in this study due to its

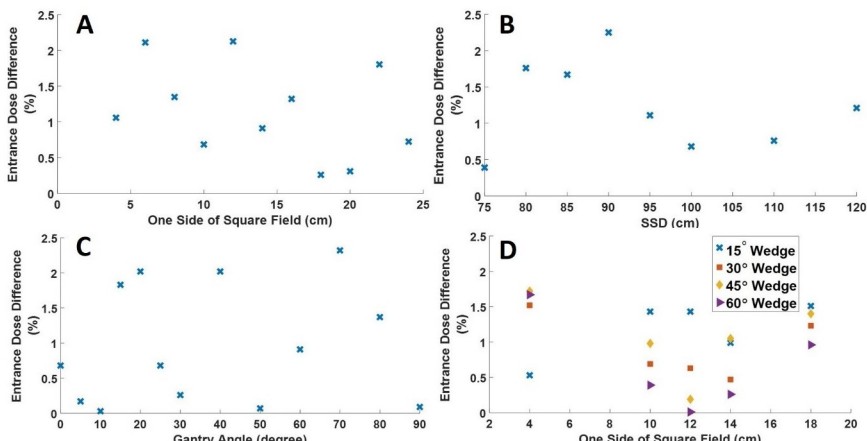

**Fig 8. Effects of irradiation parameters on the estimations of entrance dose.** The results were obtained with the same reference dose of 200 cGy. A: Shows the effect of field size, B: Shows the effect of SSD, C: Shows the effect of gantry angle, D: Shows the effect of wedge angle.

tissue-equivalent properties and ease of handling. Dose measurements using EBT2 film are affected by contaminating electrons generated in the air and collimator head of the treatment machine. The production of these electrons is influenced by factors such as photon energy, beam shaper, treatment field and SSD. In this study, a new configuration of the film setup was introduced such that the film is within an encapsulation material (made of Perspex). This new film setup minimizes the effects of the contaminating electrons on film dose estimation. How this configuration affects dose estimations has not yet been investigated. The general aim of this research was to characterize GafChromic EBT2 film dose measurements using a tissue-equivalent water phantom for a Theratron® Equinox Cobalt-60 Teletherapy Machine.

## Post-irradiation optical density growth

Post-irradiation optical density growth (coloration) is the process whereby a film continues to increase in optical density (or darken) after irradiation has ceased. As shown in Fig 3, exposing the film to ionization radiation caused an initial increase in the optical density; however, the change in optical density stabilized with time. The optical density growth for low doses of radiation stabilized at a shorter time than for high doses of radiation. After 24 hours, there were no changes in the optical density of all the irradiated films. The result of this analysis is consistent with that of previous studies [51, 52]. In a comprehensive analysis of the Gafchromic EBT2 radiochromic film by Andres, Del Castillo [40] using a 6 MV beam from a Varian 21EX LINAC (Linear Accelerator) equipped with a Millennium 80 leaf MLC and an Epson 10000XL flatbed scanner, they observed that stabilization of an irradiated film occurs earlier at low doses. Aland, Kairn [51] evaluated the Gafchromic EBT2 film dosimetry system for radiotherapy quality assurance. In their study, the films were irradiated to absorbed doses of 100 cGy, 200 cGy and 300 cGy, by a Varian 21iX linear accelerator (Varian Medical Systems, Palo Alto, USA) with an energy of 6 MV photon beam and the film readings read with a scanner. They concluded that after 24 hours of irradiation, the film's reading for the three doses of radiation stabilized with time.

## Dose uncertainty budget

From the results of this study (using Fig 5), the uncertainty in the absorbed dose estimation was ±1.79%. The uncertainty estimated in this study is within the range of uncertainty results determined from previous research [43]. Marroquin, Herrera Gonzalez [43] evaluated the uncertainty associated with the EBT3 film dosimetry system utilizing net optical density. The EBT3 film's composition and thickness of the sensitive layer are the same as those of EBT2 films. However, a matte polyester layer was added to the configuration of EBT3 film to prevent the formation of Newton's rings. Compared to EBT2 films, the symmetrical design of EBT3 allows the user to eliminate side-orientation dependence. From the analysis of the response of the radiochromic film (net optical density) and the fitting of the experimental data to a potential function, Marroquin, Herrera Gonzalez [43] observed an uncertainty of 2.6%, 4.3%, and 4.1% for the red, green, and blue channels, respectively of an Epson Perfection V750 desktop flatbed scanner. Thermoluminescent dosimeters (TLDs) are common *in vivo* dosimeters used in radiotherapy centers [53–55]. Ferguson, Lambert [53] commissioned and calibrated an automated TLD facility for measuring exit doses in external beam radiotherapy. Under their calibration conditions, the uncertainty in a single TLD measurement was approximately ±2%. The comparable uncertainty measure between TLD and EBT2 film (as determined in this study) suggests that EBT2 can be used in radiotherapy centers to complement the use of TLD for *in vivo* dosimetry.

## Assessing the effects of irradiation parameters

From Fig 6A, as the field size increased, there was a general increase in ND of the unencapsulated film compared to the encapsulated film. The result of this analysis (using just the unencapsulated film) is consistent with that of previous studies [56, 57]. Jong, Wong [57] characterized a Metal Oxide Semiconductor Field Effect Transistor (MOSFET) detector for *in vivo* skin dose measurement during megavoltage radiation therapy and compared the results with Markus ionization chamber and Gafchromic EBT2 film measurements. They observed that the relative dose absorbed by the Gafchromic film increased from 0.3 to 1.7 when the field size ranged from 1 cm x 1 cm to 40 cm x 40 cm.

As SSD increased, the optical density of the unencapsulated and encapsulated films decreased (Fig 6B). The changes in the optical density of EBT2 with variations in the SSD are consistent with findings by Jong, Wong [57]. In their study, they assessed the SSD dependence and dose rate dependence of EBT2 film and observed that as SSD and dose rate increased, the optical density of the film decreased.

As shown in Fig 6C, the response of the encapsulated film with gantry angle was approximately constant. However, increasing the gantry angle increased the optical density measurements of the unencapsulated EBT2 film due to the shift of contaminating electrons onto the film's surface. Jong, Wong [57] reported an increase in optical density with gantry angle for an unencapsulated film. Their study showed that when the gantry angle ranged from 0º to 90º, a maximum absorbed dose occurred at 70. The increased film's response was attributed to the shift of the charged particle region towards the surface of the phantom. This shift had a negligible effect on the encapsulated film because the encapsulation material absorbed and prevented the contaminating charged particles from reaching the film within.

From Fig 6D, increasing the wedge angle decreased the optical density for both the unencapsulated and encapsulated film. The decrease in optical density with increasing wedge angle has been reported in previous studies [43, 54, 55]. Bilge, Ozbek [56] measured the surface dose and build-up region with different wedge filters for photon beam energies 6 and 18 MV. They observed that for both 6 and 18 MV photon energy, an increase in the wedge angle decreased the measured radiation dose. In the presence of wedge filters, the measured doses by the encapsulated films were generally higher than that of the unencapsulated films due to the increase in the average energy of the photon beam [56].

## Use of the encapsulation material

Gafchromic EBT2 film used in radiotherapy is highly influenced by contaminating electrons at high field size and gantry angle [49, 58–60]. These contaminating electrons can increase the computational cost, time, and margin of measurement errors during *in vivo* dosimetric procedures where skin dose estimation is not a primary focus. From Fig 6, the response of the encapsulated film to variations in irradiation parameters was relatively constant (for field size and gantry angle) compared to the unencapsulated film. The fixed response of the encapsulated film without contamination at high field size and gantry angles implies high measurement accuracy, low computational cost, and time for procedures where skin dose estimation is not required. However, one drawback observed with using the encapsulated film for entrance dose measurement was the increased contribution of absorbed dose to the skin (70%). The steep rise in skin dose associated with using the encapsulated film resulted from the shift in Dmax towards the skin surface.

The design, development, and use of an encapsulation material for EBT2 films are not yet studied. This study offers a preliminary insight into how using an encapsulation material for a radiochromic film can influence dosimetric outcomes. Encapsulating the EBT2 films were

adapted from similar practice using "build-up-caps" with TLDs and MOSFETs for *in vivo* dosimetry [61, 62]. Sulieman, Theodorou [61] assessed the influence of the geometrical characteristics of cylindrical caps made of various materials at various photon fields on the TL signal. They observed that caps with appropriate thickness minimized electron contamination and ensured entrance dose measurement at Dmax. Varadhan, Miller [62] examined the feasibility of using MOSFET with a brass buildup cap for *in vivo* dose measurements and compared the entrance dose measured against that predicted by the treatment-planning system (TPS). They achieved an overall accuracy within ±5% when the measured doses were compared with that predicted by TPS. From the advantages observed with using buildup caps with TLDs and MOSFETs, further studies can be carried out on the design of the encapsulation material to minimize their perturbation of the treatment field and optimize their use for EBT2 film *in vivo* dosimetry.

## Skin dose assessment

The skin dose is an important parameter considered during external photon beam radiotherapy. Previous research has determined the skin to be at risk during radiation therapy [63–65]. Epidemiologic studies have also established a relationship between radiotherapy and the induction of basal-cell carcinoma [66–69]. Accurate determination of the surface dose is a difficult but important task for ensuring the proper treatment of patients. The percentage skin dose values estimated in this study (and shown in Fig 7) are within the range of values predicted in previous studies [70–73]. In an *in vivo* dosimetric assessments for an open field 10 cm x 10 cm and SSD of 100 cm, surface doses of 30%, 29.1%, 27.8%, 29.3%, and 29.9% were determined for five telecobalt machines: Equinox-80, Elite-80, Th-780C, Th-780, and Bhabhatron-II respectively [70]. Comparatively slightly lower skin doses (~20%) for an open field of 10 cm x 10 cm and SSD of 100 cm have been reported by Thomas and Palmer [72] and Rapley [71]. From the range of skin dose values predicted in previous studies (from~20% to 30%), the percentage skin dose estimated in this study (24%) under the same irradiation conditions can be considered reasonable. Furthermore, Rani, Ayyangar [73] evaluated the skin dose for a cobalt-60 teletherapy machine and observed that for an open field of 10 cm × 10 cm at 80 cm SSD, the % skin dose was 31.98%. The percentage of skin dose reported in their study was slightly higher than that determined in this research (29.33%) under the same irradiation conditions (open field, field size and SSD).

Results from this study further showed that large field sizes and gantry angles contributed to a substantial deposition of radiation to the skin, similar to results from previous studies [57, 74]. Yadav, Yadav [74] estimated the skin dose deposition by a 6 MV LINAC for open fields and fields with beam modifiers. Although they used a higher energy photon than what was used in this study, the characteristics of the skin dose deposition for variations in irradiation parameters were similar. They observed that relatively high amounts of skin dose were present at larger field sizes and gantry angles. Their study observed no statistically significant difference in the skin dose when the SSD ranged from 80 cm to 120 cm. The behaviors of skin dose with changes in SSD were similar to that observed in this study. Their analysis showed that the inclusion of beam modifiers during radiotherapy resulted in higher skin dose than open field irradiations. Jong, Wong [57] characterized the MOSFET detector for *in vivo* skin dose measurement during megavoltage radiotherapy. As part of their study, they assessed the characteristics of EBT2 film for varying irradiation parameters and observed a higher dependency of the film on field size and gantry angle. Jong, Wong [57] also showed that increasing the wedge angle decreases the skin dose because the presence of a physical wedge hardens the photon beam by absorbing scattered radiation.

## Entrance dose assessment

*In vivo* dosimetry during radiotherapy is necessary to assure that the dose delivered to the patient corresponds to the prescribed dose, as calculated by the treatment planning system. Entrance dose measurement is one component of *in vivo* dosimetry and is defined as the dose delivered at Dmax. The International Commission on Radiation Units and Measurements (ICRU) recommends target dose uniformity within ±5% of the absorbed dose delivered to a well-defined prescription point within the target. From Fig 8, for variations in the irradiation parameters, the measured entrance doses were within 3% of the target prescribed dose to Dmax. The degree of accuracy of the entrance dose estimations in this research agrees with a similar study by Arjomandy, Tailor [75]. In the study by Arjomandy, Tailor [75], EBT2 film was assessed as a depth-dose measurement tool for radiotherapy beams over a wide range of energies and modalities. They observed that for the Cobalt-60 photon beam, the percentage depth doses measured with the EBT2 film showed an excellent agreement (within 2%) with those measured with ionization chambers.

Traditionally, TLDs and diodes have been the choice for use as an *in vivo* dosimeter due to their high accuracy in entrance dose determinations [76–81]. Evwierhurhoma, Ibitoye [76] determined the role of TLDs during in *vivo* dosimetry. Their study was part of quality control and audit in conventional radiotherapy procedures delivered with a Co-60 teletherapy machine. They showed that no significant difference existed between the prescribed dose and measured dose of the breast with a percentage deviation difference of less than 5%. Gadhi, Fatmi [81] developed an absorbed dose verification program using the diode in vivo dosimetry system for entrance dose measurements. Phantom studies carried out in their work showed that the percentage difference between measured and calculated dose for entrance setting remained within ±2%. Their analysis also showed that entrance dose estimations for patients' measurements were within ±5% (of the prescribed dose) for open fields and ±7% for wedged fields. Comparisons of the percentage dose difference between measured and calculated entrance dose estimations for the traditional methods of *in vivo* dosimetry (both TLDs and diodes) and EBT2 films suggest that these films can be used as a complementary check to TLDs and diodes *in vivo* dose measurements.

## Limitations

Gafchromic EBT2 film has an asymmetry in the configuration of layers within it. While studies on the effect of scanning the film on the "front" side versus the "back" side by Desroches, Bouchard [82] and Carrasco, Perucha [47] confirmed a significant difference in optical density measurement. Aldelaijan, Devic [83] did not observe any significant difference in optical density measurement. In this study, we scanned only the "front" sides of the films; therefore, the effects of scanning just the "back" side on the findings of this research were not determined. The *in vivo* dosimetric protocol implemented in this research considered only the film's measurement of entrance dose and the skin dose at the entrance of the photon beam. It is possible to estimate the exit dose and subsequently the skin dose at the exit side of the photon beam; however, this was outside the scope of the current study. In this research, only the central axis absorbed doses were considered. Characterization of the film was not performed for peripheral doses. The uncertainties in the dose measurements (±1.72%) were obtained under phantom irradiation conditions. In clinical practice, higher dose uncertainties than that determined in this study may be expected due to patient-related factors such as movement, shape and composition and the variation and estimation of patient size during treatment.

## Conclusion

*In vivo* dosimetry enables the measurement of the radiation dose delivered to a target volume during radiation therapy. A tissue equivalent water phantom was used to simulate the human environment. We used GafChromic EBT2 film to assess the skin dose and entrance dose. Two configurations of the film were considered, an unencapsulated film and the other a film encapsulated within a Perspex material. From the post-exposure optical density growth, 24 hours after irradiation was determined to be optimum in the measurement of the optical density of the films. In this research, the uncertainty in dose measurements ($\pm 1.72\%$) was within the limits of acceptable uncertainty in radiotherapy absorbed dose delivery of $\pm 5\%$ specified by ICRU [84]. We assessed the film's responses for varying irradiation conditions of field size, SSD, gantry angle and Wedge angle. The unencapsulated film was observed to be more influenced by the irradiation parameters than the encapsulated film. A statistically significant difference was observed when the responses of the unencapsulated and encapsulated film were compared for varying field size ($p = 0.1802$, $\alpha = 0.05$, $n = 11$) and gantry angle ($p = 0.0018$, $\alpha = 0.05$, $n = 24$). However, no statistical difference was observed when the responses of the unencapsulated and encapsulated film were compared for changes in the SSD ($p = 0.1802$, $\alpha = 0.05$, $n = 11$) and wedge angle ($p = 0.6834$, $\alpha = 0.05$, $n = 4$).

Skin dose assessment by the two film configurations at reference conditions showed a very high radiation dose deposition by the encapsulation material (70%) compared to the EBT2 film without encapsulation (24%). Due to the very high skin dose deposited by the encapsulation material, we did not consider the encapsulated film configuration in the subsequent skin dose and entrance dose quantification. The skin dose measured using the film was higher in open field configurations compared to wedged field configurations. The estimation of the entrance dose using the unencapsulated film was within 3% of the prescribed absorbed dose for the variation in irradiation parameters considered in this study.

## Supporting information

**S1 Dataset.**
(PDF)

## Author Contributions

**Conceptualization:** Daniel Akwei Addo.

**Formal analysis:** Daniel Akwei Addo.

**Methodology:** Samuel Nii Tagoe.

**Resources:** Samuel Nii Tagoe.

**Writing – review & editing:** Elsie Effah Kaufmann, Augustine Kwame Kyere.

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
