## [Decision Letter · Decision Letter 0]

5 Jan 2022

PONE-D-21-35936IMPLEMENTATION OF IN VIVO DOSIMETRY USING GAFCHROMIC EBT2 FILM FOR A THERATRON® EQUINOXPLOS ONE

Dear Dr. Addo,

Thank you for submitting your manuscript to PLOS ONE. After careful consideration, we feel that it has merit but does not fully meet PLOS ONE’s publication criteria as it currently stands. Therefore, we invite you to submit a revised version of the manuscript that addresses the points raised during the review process.

In order for the manuscript to be considered for publication, the authors must answer thoroughly to the issues raised by the reviewers. The title of the manuscript should be changed since it does not correspond to its content.

We look forward to receiving your revised manuscript.

Kind regards,

Paula Boaventura, PhD

Academic Editor

PLOS ONE

Journal Requirements:

Reviewers' comments:

Reviewer's Responses to Questions

**Comments to the Author**

1. Is the manuscript technically sound, and do the data support the conclusions?

Reviewer #1: No

Reviewer #2: Yes

2. Has the statistical analysis been performed appropriately and rigorously? 

Reviewer #1: No

Reviewer #2: Yes

3. Have the authors made all data underlying the findings in their manuscript fully available?

Reviewer #1: No

Reviewer #2: Yes

4. Is the manuscript presented in an intelligible fashion and written in standard English?

Reviewer #1: Yes

Reviewer #2: Yes

5. Review Comments to the Author

Reviewer #1: 1.The implementation of “in vivo” dosimetry method presupposes its clinical application, or demonstration of examples of its clinical application. Meanwhile, based on the data given in the manuscript, there are no examples of clinical application or description of the irradiation conditions of real patients. Therefore, the title of the article does not correspond to its content.

2.Moreover, in the case of clinical examples, information from the authors is required on the compliance with ethical standards when performing research involves humans. This is missing from the manuscript.

2.In addition, there are a number technical errors in the manuscript. The numbering of equations in the paper is violated, for example, after the equation with the number 6 the equation with the number 5 is indicated (?).

Some terms included in the equations do not have an explanation. For example, definition of the term OD1 is not given in equation 1, the term ΠCF is indicated in Equation 4 as "product of correction factors used during the clinical applications", but this term is not clarified in relation to clinical applications.

3.Moreover, estimates of uncertainties are not indicated for the quantitative values given between equations 5 and 6.

Conclusion:

As presented, the paper cannot be recommended for publication in the Journal.

Reviewer #2: The title should be changed, as it suggests results on the use of the method on real patients will be shown when it is the characterization of the methodology proposed that is being presented.

Lines 107, 151, 498. The average energy of the Co-60 beam is 1.25 MeV (mega electron-volt) and not MV. The unit “MV” is used for megavoltage beams produced by linear accelerators. MeV stands for the energy levels of the nuclear transitions or decay of Co-60 to Ni-60 and emission of the two cascade photons.

In Materials and Methods, please mention the temperature and pressure variations allowed for, or state that correction factors to correct for the response of the ionization chamber due to temperature and pressure differences on measurement day compared to the calibration day, were used.

Lines 238 and 242. A product of correction factors is considered. Which are the correction factors?

Check the figure caption in “Figure 6 – D: Wedge angle”.

Mention that results in Figure 5 were normalized to reference conditions (10x10 cm2; SSD= 100 cm; Gantry angle 0o).

Mention that Figures 6 and 7 were obtained with, or normalized to, the same reference dose.

Line 393, “dose differences” compared to what?

In “Discussion” the first sentence (line 405-406) should be revised. The objective of radiotherapy is to deliver the prescribed dose to the target volume, sparing the surrounding healthy organs and tissues as much as possible. The sentence used is very poor.

In “references” the journal titles are not mentioned.

6. PLOS authors have the option to publish the peer review history of their article (what does this mean?). If published, this will include your full peer review and any attached files.

Reviewer #1: No

Reviewer #2: No

---

## [Author Response · Author response to Decision Letter 0]

31 Jan 2022

REVIEWER 1

1. The implementation of “in vivo” dosimetry method presupposes its clinical application, or demonstration of examples of its clinical application. Meanwhile, based on the data given in the manuscript, there are no examples of clinical application or description of the irradiation conditions of real patients. Therefore, the title of the article does not correspond to its content.

Response: Thank you for the comment. Yes, you are right! The title of the article does not correspond to its content. The title of the research has been changed to “Characterization of GafChromic EBT2 film dose measurements using a tissue-equivalent water phantom for a Theratron® Equinox Cobalt-60 Teletherapy Machine”. This is indicated on line 1-3.

2. Moreover, in the case of clinical examples, information from the authors is required on the compliance with ethical standards when performing research involves humans. This is missing from the manuscript.

Response: Thank you for this feedback. The title has been modified since this study did not involve humans. Thus, no documentation on compliance with ethical standards is included in this modified manuscript.

3. In addition, there are a number technical errors in the manuscript. The numbering of equations in the paper is violated, for example, after the equation with the number 6 the equation with the number 5 is indicated (?). 

Response: Thank you. The numbering of equations has been looked at and order restored. The equation with the number 5 now precedes the equation with the number 6 and equation 7 and 8 is also corrected. This is shown at line 265, 267, 282, 313 respectively.

4. Some terms included in the equations do not have an explanation. For example, definition of the term OD1 is not given in equation 1, the term ΠCF is indicated in Equation 4 as "product of correction factors used during the clinical applications", but this term is not clarified in relation to clinical applications. 

Response: The term OD1 has already been defined in line 142, however, this has been redefined in line 174 so that it is easy to follow. Since no clinical applications were conducted in this study, ΠCF has now been indicated as "product of correction factors used during the dose estimations; and illustrated on line 246-247".

5. Moreover, estimates of uncertainties are not indicated for the quantitative values given between equations 5 and 6. 

Response: Thank you for this feedback. Estimates of uncertainties have been included and their effect on the results assessed. This is shown at line 273-275.

REVIEWER 2

1. The title should be changed, as it suggests results on the use of the method on real patients will be shown when it is the characterization of the methodology proposed that is being presented. 

Response: Thank you for the comment. Yes, you are right! The title of the article does not correspond to its content. The title of the research has been changed to “Characterization of GafChromic EBT2 film dose measurements using a tissue-equivalent water phantom for a Theratron® Equinox Cobalt-60 Teletherapy Machine”. This is indicated on line 1-3.

2. Lines 107, 151, 498. The average energy of the Co-60 beam is 1.25 MeV (mega electron-volt) and not MV. The unit “MV” is used for megavoltage beams produced by linear accelerators. MeV stands for the energy levels of the nuclear transitions or decay of Co-60 to Ni-60 and emission of the two cascade photons. 

Response: The unit of the average energy of Co-60 beam has been changed to 1.25 MeV throughout the manuscript.

3. In Materials and Methods, please mention the temperature and pressure variations allowed for, or state that correction factors to correct for the response of the ionization chamber due to temperature and pressure differences on measurement day compared to the calibration day, were used. 

Response: “Correction factors to correct for the response of the ionization chamber due to temperature and pressure differences on measurement day compared to the calibration day” text has been included to account for possible variations in the response of the ionization chamber.

4. Lines 238 and 242. A product of correction factors is considered. Which are the correction factors? 

Response: The correction factors considered have been listed at those lines and they are correction factors for field size, SSD, gantry angle and wedge angle. This is shown on line 246 and 247.

5. Check the figure caption in “Figure 6 – D: Wedge angle”. 

Response: Thank you for the comment. The figure caption has been modified and reads “Figure 6: Effects of irradiation parameters on skin dose deposition. A: Effect of Field size; B: Effect of SSD; C: Effect of Gantry angle; D: Effect of Wedge angle”.

6. Mention that results in Figure 5 were normalized to reference conditions (10x10 cm2; SSD= 100 cm; Gantry angle 0o). 

Response: Thank you. In the results section where Figure 5 is located, statements have been added to emphasize that results in Figure 5 were normalized to reference conditions (10x10 cm2; SSD= 100 cm; Gantry angle 0o).

7. Mention that Figures 6 and 7 were obtained with, or normalized to, the same reference dose. 

Response: Thank you for this feedback. Statements have been included to emphasize that Figures 6 and 7 were obtained with the same reference dose. This is indicated on line 386-387 and line 398-399 for Figure 6 and line 406-407 and line 411-412 for Figure 7.

8. Line 393, “dose differences” compared to what? 

Response: Thank you. The dose differences were determined by comparing dose estimates using the films and the prescribed dose expected to be delivered to dmax (that is 200 cGy). This has been indicated on line 406-408.

9. In “Discussion” the first sentence (line 405-406) should be revised. The objective of radiotherapy is to deliver the prescribed dose to the target volume, sparing the surrounding healthy organs and tissues as much as possible. The sentence used is very poor. 

Response: The first sentence of the “Discussion” Section has been revised and now reads “Radiation therapy involves the delivery of a prescribed dose to the target volume, sparing the surrounding healthy organs and tissues as much as possible”. This is indicated on line 420-421.

10. In “references” the journal titles are not mentioned.

Response: Thank you. The journal titles have now been included in the “references”.

---

## [Decision Letter · Decision Letter 1]

14 Mar 2022

PONE-D-21-35936R1Characterization of GafChromic EBT2 film dose measurements using a tissue-equivalent water phantom for a Theratron® equinox Cobalt-60 teletherapy machinePLOS ONE

Dear Dr. Addo,

Thank you for submitting your manuscript to PLOS ONE. After careful consideration, we feel that it has merit but does not fully meet PLOS ONE’s publication criteria as it currently stands. Therefore, we invite you to submit a revised version of the manuscript that addresses the points raised during the review process.

The authors have addressed the reviewers’ comments. Unfortunately, Reviewer 2 did not accept to review the revised version of the manuscript (MS), so I invited a new reviewer (Reviewer #3). Please address thoroughly his comments and improve the text according to standard English.

We look forward to receiving your revised manuscript.

Kind regards,

Paula Boaventura, PhD

Academic Editor

PLOS ONE

Reviewers' comments:

Reviewer's Responses to Questions

**Comments to the Author**

1. If the authors have adequately addressed your comments raised in a previous round of review and you feel that this manuscript is now acceptable for publication, you may indicate that here to bypass the “Comments to the Author” section, enter your conflict of interest statement in the “Confidential to Editor” section, and submit your "Accept" recommendation.

Reviewer #1: (No Response)

Reviewer #3: All comments have been addressed

2. Is the manuscript technically sound, and do the data support the conclusions?

Reviewer #1: Yes

Reviewer #3: Yes

3. Has the statistical analysis been performed appropriately and rigorously? 

Reviewer #1: Yes

Reviewer #3: N/A

4. Have the authors made all data underlying the findings in their manuscript fully available?

Reviewer #1: Yes

Reviewer #3: Yes

5. Is the manuscript presented in an intelligible fashion and written in standard English?

Reviewer #1: No

Reviewer #3: Yes

6. Review Comments to the Author

Reviewer #1: I am satisfied with the authors' responses to my comments. Please carefully read your manuscript again. Your text should be improved according to standard English.

Reviewer #3: Review on the PLOS ONE manuscript "Characterization of GafChromic EBT2 film dose measurements using a tissue-equivalent water phantom for a Theratron® equinox Cobalt-60 teletherapy machine”

General comments:

In this manuscript, the authors asses a film-based methodology for dosimetry using a plastic phantom with the aim to implement it for in vivo dosimetry in different configurations. In my opinion, the authors have adequately addressed the comments made in the first review cycle and substantially improved the manuscript.

However, the text still contains several typos and sentences with unclear meaning. I suggest that the authors carefully review the text and improve its readability.

Moreover, the Discussion section is incomplete and needs revision, especially the ‘Skin dose assessment’ and ‘Entrance dose assessment’ sections. A paragraph discussing the limitations and applicability of the obtained results is also missing. What can be improved in the presented methodology?

The authors should provide some more insights on references about the encapsulation setup employed in the study, if any are existent. This part of the study, although presented in results, is completely absent from the discussion. In addition, the skin dose measured in the study is around 24% while all references indicate skin dose below 20%, and this is not examined or discussed further.

An overall uncertainty analysis related to all steps of the film dosimetry protocol is currently missing. Based on literature and/or in-house estimates, the authors should provide a detailed uncertainty budget for this methodology.

The authors present an in vivo dosimetry methodology for improving entrance and skin dose measurement, facilitating EBT2 films in different configurations and with several irradiation parameters examined. Are there any other are studies supporting this experiment? Also, are there any other studies that have examined similar irradiation parameters? Have you also tried a dosimetry setup with the encapsulation configuration, is that possible?

Line by line comments:

Line 29. Suggestion: change ‘specially constructed’ to ‘custom-made’.

Line 32. The irradiation parameters examined should be mentioned

Line 35. The sentence does not accurately describe what the aim of this section was. It should contain and compare all the irradiation parameters that were investigated and clarify that only variations of the field size or the gantry angle resulted in a statistical difference between the two film configurations.

Line 37. Suggestion: change ‘skin dose quantified’ to ‘measured skin dose’

Line 57. ‘Delivery of quality’ does not make much sense, please rephrase.

Line 219. ‘was varied’ is used here as well as in other parts of the manuscript, please rephrase.

Line 232. Change ‘ion’ to ‘ionization’, for consistency reasons (line 226 has it as ionization).

Line 242. Change ‘represent’ to ‘represents’.

Line 250. Change was to were.

Line 308. Change ‘polynomial of degree three’ to ‘3rd degree polynomial’.

Lines 403-405. The sentence does not explain the purpose of the section. This should be rephrased to better indicate how the different irradiation conditions respond to various field size, gantry angle and wedge angle configurations.

Line 542-543. Results here are inconsistent to the ones presented in lines 373 and 380 in results section and line 37 in abstract section. Please review and elaborate.

7. PLOS authors have the option to publish the peer review history of their article (what does this mean?). If published, this will include your full peer review and any attached files.

Reviewer #1: No

Reviewer #3: **Yes: **Dimitrios Dellios

---

## [Author Response · Author response to Decision Letter 1]

17 Jun 2022

REVIEWER 3

 Discussion section is incomplete and needs revision, especially the ‘Skin dose assessment’ and ‘Entrance dose assessment’ sections.

Response

Thank you for the question. The sections on skin dose and entrance dose have been examined and updated on pages 28 to 29 (for the skin dose) and pages 29 to 31 (for entrance dose). The discussions on skin dose and entrance are included here as well.

“

Skin dose assessment 

The skin dose is an important parameter considered during external photon beam radiotherapy. Previous research has determined the skin to be at risk during radiation therapy [65-67]. Epidemiologic studies have also established a relationship between radiotherapy and the induction of basal‐cell carcinoma [68-71]. Accurate determination of the surface dose is a difficult but important task for ensuring the proper treatment of patients. The percentage skin dose values estimated in this study (and shown in Fig 7) are within the range of values predicted in previous studies [72-75]. In an in vivo dosimetric assessments for an open field 10 cm x 10 cm and SSD of 100 cm, surface doses of 30%, 29.1%, 27.8%, 29.3%, and 29.9% were determined for five telecobalt machines: Equinox-80, Elite-80, Th-780C, Th-780, and Bhabhatron-II respectively [72]. Comparatively slightly lower skin doses (~20%) for an open field of 10 cm x 10 cm and SSD of 100 cm have been reported by Thomas and Palmer [74] and Rapley [73]. From the range of skin dose values predicted in previous studies (from~20% to 30%), the percentage skin dose estimated in this study (24%) under the same irradiation conditions can be considered reasonable. Furthermore, Rani, Ayyangar [75] evaluated the skin dose for a cobalt-60 teletherapy machine and observed that for an open field of 10 cm × 10 cm at 80 cm SSD, the % skin dose was 31.98%. The percentage of skin dose reported in their study was slightly higher than that determined in this research (29.33%) under the same irradiation conditions (open field, field size and SSD). 

Results from this study further showed that large field sizes and gantry angles contributed to a substantial deposition of radiation to the skin, similar to results from previous studies [58, 76]. Yadav, Yadav [76] estimated the skin dose deposition by a 6 MV LINAC for open fields and fields with beam modifiers. Although they used a higher energy photon than what was used in this study, the characteristics of the skin deposition for variations in irradiation parameters were similar. They observed that relatively high amounts of skin dose were present at larger field sizes and gantry angles. Their study observed no statistically significant difference in the skin dose when the SSD ranged from 80 cm to 120 cm. The behaviors of skin dose with changes in SSD were similar to that observed in this study. Their analysis showed that the inclusion of beam modifiers during radiotherapy resulted in higher skin dose than open field irradiations. Jong, Wong [58] characterized the MOSFET detector for in vivo skin dose measurement during megavoltage radiotherapy. As part of their study, they assessed the characteristics of EBT2 film for varying irradiation parameters and observed a higher dependency of the film on field size and gantry angle. Jong, Wong [58] also showed that increasing the wedge angle decreases the skin dose because the presence of a physical wedge hardens the photon beam by absorbing scattered radiation. 

Entrance dose assessment

In vivo dosimetry during radiotherapy is necessary to assure that the dose delivered to the patient corresponds to the prescribed dose, as calculated by the treatment planning system. Entrance dose measurement is one component of in vivo dosimetry and is defined as the dose delivered at Dmax. The International Commission on Radiation Units and Measurements (ICRU) recommends target dose uniformity within ± 5% of the absorbed dose delivered to a well-defined prescription point within the target. From Fig 8, for variations in the irradiation parameters, the measured entrance doses were within 3% of the target prescribed dose to Dmax. The degree of accuracy of the entrance dose estimations in this research agrees with a similar study by Arjomandy, Tailor [77]. In the study by Arjomandy, Tailor [77], EBT2 film was assessed as a depth-dose measurement tool for radiotherapy beams over a wide range of energies and modalities. They observed that for the Cobalt-60 photon beam, the percentage depth doses measured with the EBT2 film showed an excellent agreement (within 2%) with those measured with ionization chambers.

Traditionally, TLDs and diodes have been the choice for use as an in vivo dosimeter due to their high accuracy in entrance dose determinations [78-83]. Evwierhurhoma, Ibitoye [78] determined the role of TLDs during in vivo dosimetry. Their study was part of quality control and audit in conventional radiotherapy procedures delivered with a Co-60 teletherapy machine. They showed that no significant difference existed between the prescribed dose and measured dose of the breast with a percentage deviation difference of less than 5%. Gadhi, Fatmi [83] developed an absorbed dose verification program using the diode in vivo dosimetry system for entrance dose measurements. Phantom studies carried out in their work showed that the percentage difference between measured and calculated dose for entrance setting remained within ±2 %. Their analysis also showed that entrance dose estimations for patients’ measurements were within ±5 % (of the prescribed dose) for open fields and ±7 % for wedged fields. Comparisons of the percentage dose difference between measured and calculated entrance dose estimations for the traditional methods of in vivo dosimetry (both TLDs and diodes) and EBT2 films suggest that these films can be used as a complementary check to TLDs and diodes in vivo dose measurements. 

”

 A paragraph discussing the limitations and applicability of the obtained results is also missing. What can be improved in the presented methodology?

Response

Thank you for the question. This has been examined and updated on page 31. The write-up for the limitations section of the study is included here.

“

Limitations

Gafchromic EBT2 film has an asymmetry in the configuration of layers within it. While studies on the effect of scanning the film on the “front” side versus the “back” side by Desroches, Bouchard [84] and Carrasco, Perucha [47] confirmed a significant difference in optical density measurement. Aldelaijan, Devic [85] did not observe any significant difference in optical density measurement. In this study, we scanned only the “front” sides of the films; therefore, the effects of scanning just the “back” side on the findings of this research were not determined. The in vivo dosimetric protocol implemented in this research considered only the film’s measurement of entrance dose and the skin dose at the entrance of the photon beam. It is possible to estimate the exit dose and subsequently the skin dose at the exit side of the photon beam; however, this was outside the scope of the current study. In this research, only the central axis absorbed doses were considered. Characterization of the film was not performed for peripheral doses. The uncertainties in the dose measurements (± 1.72%) were obtained under phantom irradiation conditions. In clinical practice, higher dose uncertainties than that determined in this study may be expected due to patient-related factors such as movement, shape and composition and the variation and estimation of patient size during treatment.

”

 The authors should provide some more insights on references about the encapsulation setup employed in the study, if any are existent. This part of the study, although presented in results, is completely absent from the discussion.

Response

Thank you for the question. This has been examined and updated on page 26 to 27. The write-up for the encapsulation material section of the study is included here.

“

Use of the encapsulation material

Gafchromic EBT2 film used in radiotherapy is highly influenced by contaminating electrons at high field size and gantry angle [59-62]. These contaminating electrons can increase the computational cost, time, and margin of measurement errors during in vivo dosimetric procedures where skin dose estimation is not a primary focus. From Fig 6, the response of the encapsulated film to variations in irradiation parameters was relatively constant (for field size and gantry angle) compared to the unencapsulated film. The fixed response of the encapsulated film without contamination at high field size and gantry angles implies high measurement accuracy, low computational cost, and time for procedures where skin dose estimation is not required. However, one drawback observed with using the encapsulated film for entrance dose measurement was the increased contribution of absorbed dose to the skin (70%). The steep rise in skin dose associated with using the encapsulated film resulted from the shift in Dmax towards the skin surface. 

The design, development, and use of an encapsulation material for EBT2 films are not yet studied. This study offers a preliminary insight into how using an encapsulation material for a radiochromic film can influence dosimetric outcomes. Encapsulating the EBT2 films were adapted from similar practice using "build-up-caps" with TLDs and MOSFETs for in vivo dosimetry [63, 64]. Sulieman, Theodorou [63] assessed the influence of the geometrical characteristics of cylindrical caps made of various materials at various photon fields on the TL signal. They observed that caps with appropriate thickness minimized electron contamination and ensured entrance dose measurement at Dmax. Varadhan, Miller [64] examined the feasibility of using MOSFET with a brass buildup cap for in vivo dose measurements and compared the entrance dose measured against that predicted by the treatment‐planning system (TPS). They achieved an overall accuracy within ±5% when the measured doses were compared with that predicted by TPS. From the advantages observed with using buildup caps with TLDs and MOSFETs, further studies can be carried out on the design of the encapsulation material to minimize their perturbation of the treatment field and optimize their use for EBT2 film in vivo dosimetry.

”

 In addition, the skin dose measured in the study is around 24% while all references indicate skin dose below 20%, and this is not examined or discussed further.

Response

Thank you for the question. This has been examined and updated on page 28 to 29. The write-up for the limitations section of the study is included here.

“

Skin dose assessment 

The skin dose is an important parameter considered during external photon beam radiotherapy. Previous research has determined the skin to be at risk during radiation therapy [65-67]. Epidemiologic studies have also established a relationship between radiotherapy and the induction of basal‐cell carcinoma [68-71]. Accurate determination of the surface dose is a difficult but important task for ensuring the proper treatment of patients. The percentage skin dose values estimated in this study (and shown in Fig 7) are within the range of values predicted in previous studies [72-75]. In an in vivo dosimetric assessments for an open field 10 cm x 10 cm and SSD of 100 cm, surface doses of 30%, 29.1%, 27.8%, 29.3%, and 29.9% were determined for five telecobalt machines: Equinox-80, Elite-80, Th-780C, Th-780, and Bhabhatron-II respectively [72]. Comparatively slightly lower skin doses (~20%) for an open field of 10 cm x 10 cm and SSD of 100 cm have been reported by Thomas and Palmer [74] and Rapley [73]. From the range of skin dose values predicted in previous studies (from~20% to 30%), the percentage skin dose estimated in this study (24%) under the same irradiation conditions can be considered reasonable. Furthermore, Rani, Ayyangar [75] evaluated the skin dose for a cobalt-60 teletherapy machine and observed that for an open field of 10 cm × 10 cm at 80 cm SSD, the % skin dose was 31.98%. The percentage of skin dose reported in their study was slightly higher than that determined in this research (29.33%) under the same irradiation conditions (open field, field size and SSD). 

Results from this study further showed that large field sizes and gantry angles contributed to a substantial deposition of radiation to the skin, similar to results from previous studies [58, 76]. Yadav, Yadav [76] estimated the skin dose deposition by a 6 MV LINAC for open fields and fields with beam modifiers. Although they used a higher energy photon than what was used in this study, the characteristics of the skin deposition for variations in irradiation parameters were similar. They observed that relatively high amounts of skin dose were present at larger field sizes and gantry angles. Their study observed no statistically significant difference in the skin dose when the SSD ranged from 80 cm to 120 cm. The behaviors of skin dose with changes in SSD were similar to that observed in this study. Their analysis showed that the inclusion of beam modifiers during radiotherapy resulted in higher skin dose than open field irradiations. Jong, Wong [58] characterized the MOSFET detector for in vivo skin dose measurement during megavoltage radiotherapy. As part of their study, they assessed the characteristics of EBT2 film for varying irradiation parameters and observed a higher dependency of the film on field size and gantry angle. Jong, Wong [58] also showed that increasing the wedge angle decreases the skin dose because the presence of a physical wedge hardens the photon beam by absorbing scattered radiation. 

”

 An overall uncertainty analysis related to all steps of the film dosimetry protocol is currently missing. Based on literature and/or in-house estimates, the authors should provide a detailed uncertainty budget for this methodology.

Response

Thank you for the question. This has been examined and updated on page 9 to 10; 17; 24. The write-up for the encapsulation material section of the study is included here.

“

Dose uncertainty budget

The dose uncertainties carried out in this study were calculated by error propagation as used in previous studies [46-48]. Two sources of uncertainties considered during the generation of the calibration curve were: the experimental and fitting. We assumed the scanning of the films to be the source of the experimental uncertainties, while the curve fitting uncertainties were associated with the accuracy of the curve fitting process.

Equation 2 represents a potential equation for the calibration curve:

Dose=a〖OD〗^3+b〖OD〗^2+cOD+d ………. Equation 2 

Where:

OD is the measured optical density; Dose is the estimated dose; a, b, c and d are the coefficients. 

The final experimental dose uncertainty was calculated by applying Equation 3:

δ_(e-dose)= (3a〖OD〗^2+2bOD+OD)*δ_OD ………. Equation 3

where: δe-dose, is the experimental dose uncertainty; OD, is the optical density measurement; δ_OD, is the standard deviation of the densitometer. Since the digital display of the densitometer is limited to 2 decimal places, the standard deviation of the densitometer was approximated to 0.01.

The curve fitting uncertainty was calculated by applying Equation 4: 

δ_(f-dose)= √(δ_a^2 〖OD〗^6+δ_b^2 〖OD〗^4+〖δ_c^2 OD〗^2+δ_d^2 )) ………. Equation 4

Where: δ_(f-dose) is the fitting uncertainty; δ_a,δ_b,δ_c,& δ_d are the standard deviation of the fitting parameters. The standard deviation of the fitting parameters was calculated using the non-linear model fit function in MATLAB (The Math Works, Inc. MATLAB. Version 2020a).

Finally, the total dose uncertainty was calculated using the Equation 5:

δ_dose= √(〖(δ〗_(e-dose)^2+δ_(f-dose)^2 )) ………. Equation 5

Fig 5 shows a plot of dose uncertainties against estimated absorbed doses using EBT2 films. From Fig 5, increasing the absorbed radiation dose increased the absorbed dose uncertainty. The minimum dose uncertainty recorded was 0.12% (at 50cGy), while the maximum was 1.72% (at 800 cGy).

Fig 5: Plot of total dose uncertainty against absorbed dose for EBT2 film. The percentage uncertainty in absorbed dose estimations increased as the dose to the film increased.

From the results of this study (using Fig 5), the uncertainty in the absorbed dose estimation was ±1.79%. The uncertainty estimated in this study is within the range of uncertainty results determined from previous research [53]. Marroquin, Herrera Gonzalez [53] evaluated the uncertainty associated with the EBT3 film dosimetry system utilizing net optical density. The EBT3 film's composition and thickness of the sensitive layer are the same as those of EBT2 films. However, a matte polyester layer was added to the configuration of EBT3 film to prevent the formation of Newton's rings. Compared to EBT2 films, the symmetrical design of EBT3 allows the user to eliminate side‐orientation dependence. From the analysis of the response of the radiochromic film (net optical density) and the fitting of the experimental data to a potential function, Marroquin, Herrera Gonzalez [53] observed an uncertainty of 2.6%, 4.3%, and 4.1% for the red, green, and blue channels, respectively of an Epson Perfection V750 desktop flatbed scanner. Thermoluminescent dosimeters (TLDs) are common in vivo dosimeters used in radiotherapy centers [54-56]. Ferguson, Lambert [54] commissioned and calibrated an automated TLD facility for measuring exit doses in external beam radiotherapy. Under their calibration conditions, the uncertainty in a single TLD measurement was approximately ±2%. The comparable uncertainty measure between TLD and EBT2 film (as determined in this study) suggests that EBT2 can be used in radiotherapy centers to complement the use of TLD for in vivo dosimetry.

”

 The authors present an in vivo dosimetry methodology for improving entrance and skin dose measurement, facilitating EBT2 films in different configurations and with several irradiation parameters examined. 

 Are there any other are studies supporting this experiment? 

Response

In this study, we considered two configurations of the film: the unencapsulated (common usage) and encapsulated film. Some studies have assessed the use of the unencapsulated film for in vivo dosimetry. The results of those studies agree with the results of this research. These are documented in the write-up (Page 26 to 27). Currently, no study has examined in vivo dosimetry using an encapsulated film. This study offers a preliminary insight into how using an encapsulation material for a radiochromic film can influence dosimetric outcomes. Encapsulating the EBT2 films were adapted from similar practice using "build-up-caps" with TLDs and MOSFETs for in vivo dosimetry [63, 64].

 Also, are there any other studies that have examined similar irradiation parameters? 

Response

Yes, several studies have examined similar irradiation parameters but on just the unencapsulated film and these are presented on pages 23, 25 to 26.

 Have you also tried a dosimetry setup with the encapsulation configuration, is that possible?

Response

Yes, we tried a dosimetry setup with the encapsulation configuration for skin dose assessment as well as assessment of the effects of irradiation parameters on the encapsulated film’s response. The presence of the encapsulation material increased the skin dose to about three times the amount that would have been deposited using an unencapsulated film. Because of the high skin dose deposition observed with the encapsulation material, we did not use this configuration for further dosimetric assessment. However, we proposed that extensive studies should be carried out with the encapsulation material to optimize its use for in vivo dosimetry. 

A detailed assessment of the use of encapsulated film configuration for in vivo dosimetry is presented on Pages 26 to 27 and shown below.

“

Use of the encapsulation material

Gafchromic EBT2 film used in radiotherapy is highly influenced by contaminating electrons at high field size and gantry angle [59-62]. These contaminating electrons can increase the computational cost, time, and margin of measurement errors during in vivo dosimetric procedures where skin dose estimation is not a primary focus. From Fig 6, the response of the encapsulated film to variations in irradiation parameters was relatively constant (for field size and gantry angle) compared to the unencapsulated film. The fixed response of the encapsulated film without contamination at high field size and gantry angles implies high measurement accuracy, low computational cost, and time for procedures where skin dose estimation is not required. However, one drawback observed with using the encapsulated film for entrance dose measurement was the increased contribution of absorbed dose to the skin (70%). The steep rise in skin dose associated with using the encapsulated film resulted from the shift in Dmax towards the skin surface. 

The design, development, and use of an encapsulation material for EBT2 films are not yet studied. This study offers a preliminary insight into how using an encapsulation material for a radiochromic film can influence dosimetric outcomes. Encapsulating the EBT2 films were adapted from similar practice using "build-up-caps" with TLDs and MOSFETs for in vivo dosimetry [63, 64]. Sulieman, Theodorou [63] assessed the influence of the geometrical characteristics of cylindrical caps made of various materials at various photon fields on the TL signal. They observed that caps with appropriate thickness minimized electron contamination and ensured entrance dose measurement at Dmax. Varadhan, Miller [64] examined the feasibility of using MOSFET with a brass buildup cap for in vivo dose measurements and compared the entrance dose measured against that predicted by the treatment‐planning system (TPS). They achieved an overall accuracy within ±5% when the measured doses were compared with that predicted by TPS. From the advantages observed with using buildup caps with TLDs and MOSFETs, further studies can be carried out on the design of the encapsulation material to minimize their perturbation of the treatment field and optimize their use for EBT2 film in vivo dosimetry.

”

 Line 29. Suggestion: change ‘specially constructed’ to ‘custom-made’.

 Line 32. The irradiation parameters examined should be mentioned

 Line 35. The sentence does not accurately describe what the aim of this section was. It should contain and compare all the irradiation parameters that were investigated and clarify that only variations of the field size or the gantry angle resulted in a statistical difference between the two film configurations.

 Line 37. Suggestion: change ‘skin dose quantified’ to ‘measured skin dose’

 Line 57. ‘Delivery of quality’ does not make much sense, please rephrase.

 Line 219. ‘was varied’ is used here as well as in other parts of the manuscript, please rephrase.

 Line 232. Change ‘ion’ to ‘ionization’, for consistency reasons (line 226 has it as ionization).

 Line 242. Change ‘represent’ to ‘represents’.

 Line 250. Change was to were.

 Line 308. Change ‘polynomial of degree three’ to ‘3rd degree polynomial’.

 Lines 403-405. The sentence does not explain the purpose of the section. This should be rephrased to better indicate how the different irradiation conditions respond to various field size, gantry angle and wedge angle configurations.

 Line 542-543. Results here are inconsistent to the ones presented in lines 373 and 380 in results section and line 37 in abstract section. Please review and elaborate.

Response

Thank you for your comments.

The write-up has been updated to reflect the recommended changes addressed from points 7 to 18.

---

## [Editor Report · Decision Letter 2]

22 Jun 2022

Characterization of GafChromic EBT2 film dose measurements using a tissue-equivalent water phantom for a Theratron® equinox Cobalt-60 teletherapy machine

PONE-D-21-35936R2

Dear Dr. Addo,

We’re pleased to inform you that your manuscript has been judged scientifically suitable for publication and will be formally accepted for publication once it meets all outstanding technical requirements.

Kind regards,

Paula Boaventura, PhD

Academic Editor

PLOS ONE
---

## [Editor Report · Acceptance letter]

11 Aug 2022

PONE-D-21-35936R2 

Characterization of GafChromic EBT2 film dose measurements using a tissue-equivalent water phantom for a Theratron® equinox Cobalt-60 teletherapy machine 

Dear Dr. Addo:

I'm pleased to inform you that your manuscript has been deemed suitable for publication in PLOS ONE. Congratulations! Your manuscript is now with our production department. 

Kind regards, 

on behalf of

Dr. Paula Boaventura 

Academic Editor

PLOS ONE